# Heat Treatment at an Early Age Has Effects on the Resistance to Chronic Heat Stress on Broilers

**DOI:** 10.3390/ani9121022

**Published:** 2019-11-23

**Authors:** Darae Kang, JinRyong Park, KwanSeob Shim

**Affiliations:** Department of Animal Biotechnology, Jeonbuk National University, Jeonju 54896, Korea; kangdr92@gmail.com (D.K.); wlsfyd1321@naver.com (J.P.)

**Keywords:** chronic heat stress, heat shock protein, broiler, resistance

## Abstract

**Simple Summary:**

In this study, we investigated early heat exposure to a chronic heat-stressed group, of which has effects on growth performance, liver-specific enzymes (GOT; glutamic oxalacetic transaminase, and GPT; glutamic pyruvate transaminase), neuro (dopamine and serotonin) and stress (corticosterone) hormones, and the expression of HSPs (heat shock proteins), HSFs (heat shock factors), and pro-inflammatory cytokines. Chronic heat stress and early heat exposed groups reduced body weight, feed intake, and GOT, GPT, serotonin in the serum, and the protein expression of HSPs in liver tissue. At the same time, HSP70, HSP27, TNF-α, IFNG, and IL-6 gene expression were up-regulated in the early heat exposed group. According to study results regarding the broilers, early heat exposure has no effects on growth performance, physiology, and HSP protein expressions.

**Abstract:**

This study was conducted to investigate the effects of early heat conditioning on growth performance, liver-specific enzymes (GOT and GPT), neuro-hormones (dopamine and serotonin), stress hormones (corticosterone), and the expression of HSPs (heat shock proteins), HSFs (heat shock factors), and pro-inflammatory cytokines under chronic high temperature. Broilers were raised with commercial feed and supplied with water ad libitum under conventional temperature. We separated the broilers into three groups: the control without any heat exposure (C), chronic heat-stressed group (CH), and early and chronic heat-stressed group (HH). At 5 days of age, the HH group was exposed to high temperatures (40 °C for 24 h), while the remaining groups were raised at a standard temperature. Between days 6 and 20, all three groups were kept under optimal temperature. From 21 to 35 days, the two heat-stressed groups (CH and HH) were exposed to 35 °C. Groups exposed to high temperature (CH and HH) showed significantly lower body weight and feed intake compared to the control. GOT and GPT were lower expressed in the CH and HH groups than the control group. In addition, the protein expressions of HSPs were down-regulated by chronic heat stress (CH and HH groups). The gene expressions of HSP60 and HSF3 were significantly down-regulated in the CH and HH groups, while HSP70 and HSP27 genes were up-regulated only in the HH group compared with the control group. The expression of pro-inflammatory cytokine genes was significantly up-regulated in the HH group compared with the control and CH groups. Thus, exposure of early Heat stress (HS) to broilers may affect the inflammatory response; however, early heat exposure did not have a positive effect on chronic HS of liver enzymes and heat shock protein expression.

## 1. Introduction

Heat stress (HS), caused by climate change, negatively affects animal performance, especially chickens during the summer season. The longer the period of high temperature, the more severe and persistent is the effect of HS on the chickens, which has a negative economic impact. HS has adverse effects on poultry production in summer days, suggesting the need for the development of high temperature resilience in broilers. Few studies have investigated the improved heat resistance to HS in chickens via feed restriction [1], supplementation of electrolytes, vitamin C, betaine and selenium with feed or water [2,3,4,5]. Most of the previous studies reported that nutritional supplementation improved the heat tolerance and productivity, and regulated the oxidative damage markers and hormonal changes [1,2,3,4,5].

Early heat treatment also induced heat resistance in poultry. Previous studies investigated the effects of late chronic HS under pre- or post-hatching heat conditions. Heat treatment prior to hatching (39.5 °C/12 h/day, from day 7 to 16 of incubation) and post-hatching (35–37.8 °C, 24 h, on day 5 after hatching) improved heat resistance after late HS [6,7]. Zaboli et al. [7] and Yalcin et al. [8] reported that early heat conditions after hatching had a positive effect in inhibiting the growth decrease caused by late chronic HS. However, initial heat exposure had no effect on mortality, weight gain, or feed efficiency of broilers under chronic HS [9]. There were no positive effects on the hens’ ability to adapt to heat [10]. Currently, the effect of early heat treatment on late chronic HS is still disputed.

The expression of heat shock proteins (HSPs) in the liver is rapid and is dramatically increased in animals exposed to environmental and mental stress, resulting in cellular protection by repair or unfolding proteins, and indicating the stress levels. In broilers, acute HS elevated the HSP70 level in the bursa of Fabricius, thymus, spleen, and liver [11]. As well, the expression of HSP70 and HSP90 was upregulated by chronic HS in the liver [12,13]. Under early heat conditions, the HSP concentrations showed varying results. Vinoth et al. [14] reported that early heat treatment during incubation increased the heat adaptability by inhibiting the expression of HSP90a, HSP70, and HSP60 by chronic HS after hatching. Other researchers [15] showed that the high levels of HSP70 expression in groups exposed to early heat treatment after chronic HS may reduce the effects of heat stress. Another study [1] concluded that early heat treatment had no effect on late HS because no significant variation in the levels of HSP70 was observed between groups exposes to early and delayed heat treatment. As described above, studies investigating HSP expression following initial heat treatment showed different results.

Few studies investigated the effect of early heat conditions on chronic HS. The studies usually analyzed the growth performance, mortality rate, oxidative stress, heat shock proteins, and inflammatory factors, and the results were not consistent. Therefore, the purpose of this study was to investigate the activity of liver specific enzymes, the levels of neuro-hormones and stress hormones as well as the expression of HSPs, HSF (heat shock factors), and pro-inflammatory cytokines under chronic heat stress following early heat exposure.

## 2. Materials and Methods

All animal experiments were approved by the Animal Ethics Committee of Jeonbuk National University (CBNU2018-097), Republic of Korea.

### 2.1. Animals and Treatments

One-day-old Ross broilers (Dong-woo, Iksan, Korea) (*n* = 144) were raised separately under different temperature of control. Different chicks in groups of 12 were randomly allocated to three treatments, and four replicates. The birds were reared in cages (130 cm × 100 cm × 62 cm, length × width × height) with an open ceiling. The experimental schedule of the temperature is shown in Figure 1. The control group (C) was maintained under normal temperature conditions without any heat exposure. The temperature and humidity were 34 °C and 50% during the first week, respectively. The weekly temperature was reduced by 2 °C, and was 24 °C at week 5. The group exposed to chronic heat stress (CH) was raised in the same conditioned room with control until day 21. The 21-day-old chicks were exposed to heat environment (35 ± 2 °C) until day 35. The group exposed to early heat (HH) was controlled similar to the CH group, and also exposed to early heat at 40 °C for 24 h on day 5. All birds were fed commercial diets (Table 1). The feed and water were provided ad libitum with a 12 h light and 12 h dark cycle. Feed intake of each 4 replication and body weight of individual birds were measured on days 0, 7, 14, 21, 28, and 35. Cumulative feed intake and feed conversion ratio (FCR) were calculated on a cage basis. After heat exposure on day 35, 10 animals in each group were randomly selected, and their blood samples were collected from the wing vein, and sacrificed. The liver tissues were taken, immediately frozen in liquid nitrogen, and stored until analysis at −80 °C.

### 2.2. Determination of Serum GOT and GPT

Blood was centrifuged, and serum was collected. The serum GOT (glutamic oxalacetic transaminase) and GPT (glutamic pyruvate transaminase) levels were measured using a quantitative analysis kit (Asan set GOT and GPT, Giheung, Korea) via Reitman–Frankel method. For the analysis of GOT or GPT, 1 mL of reagent (l-aspartic acid and α-ketoglutaric acid mixture for GOT; dl-alanine and α-ketoglutaric acid for GPT) was incubated at 37 °C for 5 min, followed by the addition of 0.2 mL of serum, vortexed and left at 37 °C for 60 min. It was mixed with 2, 4-dinitrophenylhydrazine reagent and after 20 min, supplemented with sodium hydroxide and allowed to stand for 10 min. The absorbance was measured at 505 nm within 60 min using a micro-reader (Thermo, Waltham, MA, USA). 

### 2.3. Determination of Serum Dopamine, Serotonin, and Corticosterone Levels

Dopamine, serotonin, and corticosterone levels in the serum were analyzed via LC-MS/MS according to the slightly modified method of Marwah et al. [16]. Serum and methanol (1:9, v:v) were vortexed and incubated at −20 °C for 1 h. All the samples were centrifuged and the supernatant was transferred to a 2 mL vial for injection. The samples were analyzed using LC-MS/MS, and the operating conditions are shown in Table 2.

### 2.4. Determination of HSPs, HOP, and HIP by Western Blot

The expression of HSP90, HSP70, HSP60, HSP47, HSP40, HOP (hsp70-hsp90 organizing protein), and HIP (hsp70 interacting protein) was measured in liver tissue using a Western blot [17]. Liver tissue was powdered under liquid nitrogen.

For sample preparation, the powdered sample was mixed with RIPA buffer (radio immune precipitation assay buffer) and centrifuged (12,000 rpm, 10 min, 4 °C). The supernatant was collected. The total protein was quantified using DC kit (Bio-Rad, Hercules, CA, USA), and the absorbance was measured at 750 nm. The supernatant and sample buffer mixture (1:1, v:v) was incubated at 95 °C for 5 min in a water-bath. The samples were separated with 12% acrylamide gel, transferred to PVDF (poly vinylidene fluoride) membrane, and blocked using 5% skim milk with TBST buffer. Each membrane was incubated with a primary antibody overnight at 4 °C, and a secondary antibody for 1.5 h at room temperature. The primary antibodies such as HSP90 (1:1250, ab13492, Abcam, Cambridge, UK), HSP70 (1:2500, ADI-SPA-820, Enzo, San Diego, CA, USA), HSP60 (1:2500, ADI-SPA-806, Enzo, San Diego, CA, USA), HSP40 (1:1250, ADI-SPA-400, Enzo, Farmingdale, NY, USA), HOP (1:2500, ADI-SRA-1500, Enzo, Victoria, BC, Canada), HIP (1:1250, MA3-413, Thermo, Golden, CO, USA), and GAPDH (1:2500, sc-47724, Santa Cruz, CA, USA) were diluted with 3% skim milk. The secondary antibodies against each of the primary antibodies (dilution ratio was twice that of each antibody) were diluted with 5% skim milk. Blots were reacted using the SuperSignal chemiluminescent substrates (ThermoFisher, Waltham, MA, USA) following the manufacturer’s instructions, and the protein bands were imaged and measured their density values using the iBright Imaging System (Thermo, San Jose, CA, USA). The protein expression was normalized by GAPDH.

### 2.5. Gene Expression by Real Time qPCR

Total RNA was extracted from liver tissue using the nucleic acid extraction kit (AccuPrep, Universal RNA extraction kit, Bioneer), and quantified using with the Thermo Scientific™ μDrop™ Plate (Thermo Scientific, USA) and Multiscan GO (Thermo Scientific, USA) at an absorbance ratio of 260/280. Complementary DNA (cDNA) was synthesized using Accupower cyclescript RT premix (Bioneer, Korea) according to the manufacturer’s instructions. RT-qPCR was performed using the CFX Real-Time System (C1000 Thermal cycler, Bio-Rad, Hercules, CA, USA). The RT-qPCR primers including HSPs (HSP108, HSP90, HSP70, HSP60, HSP47, HSP40, HSP27), ST13 (suppression of tumorigenicity 13), HSFS (HSF1, HSF2, HSF3), TNF-α (tumor necrosis factor alpha), CRYAB (alpha crystallin B), IFNG (interferon gamma), IL-6, IL-1B, and GAPDH (glyceraldehyde 3-phosphate dehydrogenase) were designed (Table 3). The cDNA amplification was performed using SsoFast EvaGreen Supermix (Bio-Rad) under the following conditions: an enzyme activation step at 95 °C for 2 min, followed by denaturation and annealing/extension steps involving 40 cycles each at 95 °C for 5 s and at 60 °C (or each primer’s annealing temperature) for 5 s. The gene expression was normalized with GAPDH which is a housekeeping gene. The related gene expression was calculated via 2^−ΔΔCT^ method [18].

### 2.6. Statistical Analysis

The SAS 9.4 software program (SAS Institute Inc, USA) was used for statistical analysis of all data, via one-way analysis of variance (ANOVA). Statistical differences among the groups were grouped by Duncan’s new multiple range test. All data were expressed as means ± standard error, and the significance level was set at *p* < 0.05.

## 3. Results

### 3.1. Growth Performance

The results of body weight gain and feed intake of chicks are shown in Table 4. Exposure of 5-day-old broilers to early heat (HH) showed no effect on body weight and feed intake in the first week. Between days 8 and 21, all chicks were raised at optimal temperature, and no significant differences in body weight, BDG, feed intake, and FCR were found among the groups. After exposure to chronic heat (21–35 days, 35 °C/24 h/day), the levels of body weight, BDG, and feed intake were significantly lower (*p* < 0.05) in the CH and HH groups than in the control. However, no differences in FCR were observed among the groups (*p* > 0.05).

### 3.2. Serum GOT and GPT Assay

Figure 2 shows the serum GOT and GPT concentrations in experimental groups. The concentrations of both GOT and GOT were significantly lower (*p* < 0.05) in CH and HH groups than in the control.

### 3.3. Analysis of Serum Dopamine, Serotonin, and Corticosterone Concentrations

The levels of dopamine, serotonin, and corticosterone are shown in Figure 3. The levels of dopamine and corticosterone were not significantly altered among the groups. The serotonin level was significantly lower (*p* < 0.05) in the HH group than in the control. Intermediate value was found in CH group (*p* > 0.05).

### 3.4. Expression of HSPs and Related Proteins by Western Blot

Figure 4 shows the results of Western blot analysis of HSPs including HSP90, HSP70, HSP60, HSP47, HSP40, and HSP related proteins namely HOP and HIP. The expression of HSP70 and HSP60 was significantly down regulated in CH and HH groups than in the control. The HH group showed a significantly lower expression of HSP60 compared with the CH group. The HSP40 expression was significantly (*p* < 0.05) lower in the HH group than in the control. However, the HOP expression was significantly up-regulated in the HH group than in the control and the CH group. No apparent changes in the expression of HSP related protein such as HSP90 and HIP were found among the experimental groups.

### 3.5. RT-qPCR Amplification of HSPs, HSFs, and Pro-Inflammatory Cytokine Gene Expression

We analyzed the mRNA expression of HSPs (HSP90, HSP70, HSP60, HSP47, HSP40, and HSP27), HSFs (HSF1, HSF2, and HSF3), and pro-inflammatory cytokines (TNF-α, IFNG, and IL-6) presented in Figure 5 and Figure 6. Each gene was normalized to that of a housekeeping gene (GAPDH). The expression of HSP90, HSP47, HSP40, HSF1, and HSF2 showed no significant difference (*p* > 0.05) among the experimented groups, whereas the expression of HSP70 and HSP27 gene was significantly up regulated (*p* < 0.05) in the HH group than in the control, even HSP70 was significantly higher (*p* < 0.05) than in the CH group, and HSP27 of the CH group was higher (*p* < 0.05) compared with the control. Further, the levels of HSP60 and HSF3 were significantly low (*p* < 0.05) in CH and HH groups than in the control. The levels of pro-inflammatory cytokine genes including TNF-α, IFNG, and IL-6 were significantly higher (*p* < 0.05) in the HH group than in C and CH groups.

## 4. Discussion

Low body weight gain is predictable when the chicken is heat stressed, due to increases in feed digestion, which is known to inhibit feed intake. Previous studies already reported that under chronic heat stress (37 °C/8 h/day, for 15 days), broilers showed a significantly low weight gain and feed intake compared with the control group [11]. Studies conducted by Yalcin et al. [8] showed that 3 weeks of heat treatment (32 to 35 °C/7 h/day) reduced the body weight gain in chickens. A previous study reported that in ovo injection of galactooligosaccharides could maintain the body weight during heat stress [19]. However, the CH and HH groups showed a lower body weight gain than the control group, that showed the early heat exposure have no effect on body weight (Table 4). Our results are consistent with a previous study of Zaboli et al. [7] showing that early heat control did not affect weight gain in 7-day-old chicks, that the group exposed to early heat treatment showed no significant difference in body weight gain compared with the group exposed to chronic heat stress after both groups were exposed to chronic heat. Conversely, early heat treatment reduced the body weight of 7-day-old chicks, followed by accelerated weight gain, reducing the weight difference from the control exposed to chronic HS [8]. In addition, as shown by Arjona et al. [6], early heat exposure suppressed weight loss by HS before marketing. This difference from previous studies was attributed to experimental conditions and differences in chicken breed. Most previous studies reported chronic HS with cyclic conditions and recovery items for chickens. In this study, under treatment with continuous heat, early heat exposure may not affect late heat treatment.

The serum GOT and GPT levels are indicators of liver damage because they are released into the blood when the liver is damaged [20]. Agrawal and Gupta [21] reported that serum GPT and GOT in heat-stressed chickens were significantly increased compared with the control, indicating liver damage. Chronic HS (30–32 °C, ranging from 20 weeks to 27, 43, 55, and 68 weeks) also stimulated plasma levels of GOT and GPT [22]. By contrast, previous studies of GPT levels for 4 weeks (32 °C) [23] and 8 days (33 °C) [24] following exposure to constant heat stress showed no significant difference. Similarly, Ma et al. [25] found that plasma GOT and GPT levels were significantly decreased in heat exposed duck (34 °C for 28 days). Our results also showed significantly low GOT and GPT levels in chronic heat stress (CH and HH) compared with the control (Figure 2). These results demonstrated that long-term heat stress affected enzyme synthesis and turnover, suggesting that under prolonged and severe liver damage, the concentration of enzymes in the blood declined.

Endogenous serotonin levels in the serum of the group exposed to early heat (HH) were significantly lower, compared with the control group (Figure 3). Exposure to chronic heat stress at 35 °C for 14 days (CH) showed no significant changes in dopamine, serotonin, and corticosterone levels, compared with the control group (Figure 3). These findings are consistent with earlier reports, suggesting that serotonin levels were not different from the control during chronic heat exposure (constantly at 34 °C for 5 weeks), whereas the group exposed to acute stress (34 °C, 6 h) showed significantly low serotonin levels [26]. Daily exposure of three different rat models (4–5 weeks, 9–11 weeks, and above 6 months) to heat for 1 h at 38 °C for 21 days resulted in no significant difference in plasma corticosterone levels in all the age groups, while acute heat stress (38 °C, 4 h) resulted in significant up regulation in corticosterone levels [27]. Further, Sinha [28] reported that chronic heat stress (38 °C, 1 h daily for 21 days) does not affect the corticosterone level compared with the control group. The above studies are consistent with our results suggesting that chronic heat stress affects the serotonin levels but not dopamine or corticosterone levels. The normal stress response should reduce serotonin and increase corticosterone. However, chronic HS in the present study showed a different effect. Chronic stress may impair the activity of hypothalamic-pituitary-adrenocortical axis [29], although early heat conditions reduced damage, suggesting a decrease in serotonin secretion compared with the CH group.

This study demonstrated that chronic HS significantly down-regulated the expression of HSP70, HSP60, and HSP40 as well the mRNA expression of HSP60 and HSF3 in liver tissue (*p* < 0.05). Further, early heat treatment down-regulated HSP60 and HSP40 proteins (*p* < 0.05) (Figure 4 and Figure 5). In contrast, the levels of HOP protein and HSP70 gene were up-regulated in the group exposed to early heat stress (HH) (Figure 4 and Figure 5). HSPs are strongly induced by many types of stress, especially heat, to repair altered protein and folding caused by cellular damage [30]. According to a previous study, the colored broilers (NN and PB-2 species) subjected to chronic heat stress (35 °C, 15–42 days) showed an upregulation of HSP90 alpha, HSP60, and HSP70 genes resulting in significantly higher levels of HSP90 alpha, HSP60, and HSP70 protein expression than the control groups [14]. Previous studies demonstrated that high levels of HSPs indicated increased heat resistance in which Zulkifli et al. [31] and Tamzil et al. [32] found that HS stimulated the expression of HSP70 protein and gene expression compared with the control. In contrast, other studies [32,33,34] reported a lower level of HSP expression during heat exposure. These results may be attributed to differences in experimental conditions, environment, individuals, or animal species. 

This study showed that the mRNA expression of TNF-α, IFNG, and IL-6 in the liver was significantly upregulated in the HH group (*p* < 0.05), while the CH group showed no significant difference from the control group (Figure 6). Therefore, early heat conditions activated the transcription of cytokines but not in chronic HS. Pro-inflammatory cytokines such as TNF-α, IFNG, and IL-6 are the main indicators of fever and inflammation [35]. Oskoueian et al. [36] reported that HS (40 °C, 180 min) in chicken increased the expression of IFNG, TNF-like, and IL-1b genes and eventually produced NO and inflammatory reactions. In addition, Yun et al. [37] demonstrated that exposure to cyclic HS (32 °C/2 h/day for 7 days) led to a significantly higher TNF-α expression, resulting in inflammatory response. In contrast to the reports above, exposure to chronic HS (34 °C, 12 days) showed no significant difference in TNF-α and IL-6 gene expression [38]. Another study [39] showed that exposure to HS (38 °C/4 h/day for 21 days) significantly decreased the levels of IL-6, IFN-b, and IL-10, consistent with our results. Therefore, early heat conditions may have an effect on immune response in chickens, suggesting the ability to defend against HS.

In this study, the growth performance, liver enzymes, neuropeptides, corticosterone levels, expression of HSPs, HSFs, and pro-inflammatory cytokines after chronic HS and early heat treatment were determined. The expression of pro-inflammatory cytokine genes was upregulated in groups treated with early heat, which suggests a positive effect on the inflammatory response. However, the body weight, BDG, feed intake, GOT, GPT, and protein expression of HSPs were significantly decreased in chronic HS, which means that early heat exposure had no effects. As a result, we found that early heat conditions have no effect on HSP protein metabolism in livers damaged by chronic HS, and these findings may be used as basic data on management practices to improve the thermal resistance of broilers.

## Figures and Tables

**Figure 1 animals-09-01022-f001:**
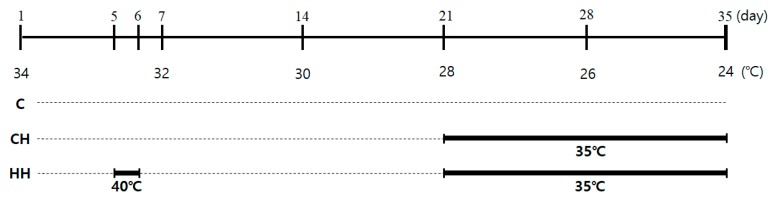
Experimental schedule of the heat exposure method. C, control; CH, chronic heat-stressed broiler; HH, early and chronic heat-stressed broiler.

**Figure 2 animals-09-01022-f002:**
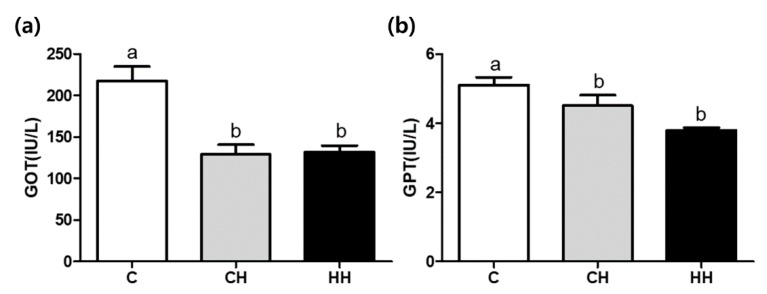
GOT (glutamic oxalacetic transaminase) and GPT (glutamic pyruvate transaminase) concentration of serum. (**a**) GOT; (**b**) GPT. C, control; CH, chronic heat-stressed broiler; HH, early and chronic heat-stressed broiler. ^a,b^ Different superscript letters are significantly different (*p* < 0.05).

**Figure 3 animals-09-01022-f003:**
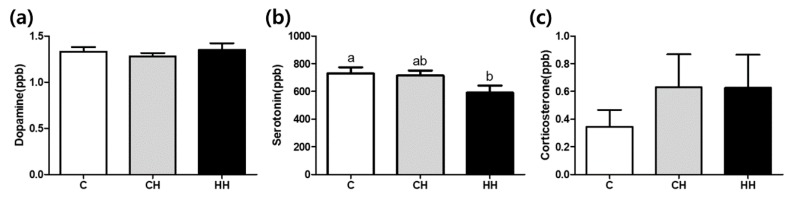
Dopamine, serotonin, and corticosterone levels of serum. (**a**) Dopamine (ppb); (**b**) serotonin (ppb); (**c**) corticosterone (ppb). C, control; CH, chronic heat-stressed broiler; HH, early and chronic heat-stressed broiler. ^a,b^ Different superscript letters are significantly different (*p* < 0.05).

**Figure 4 animals-09-01022-f004:**
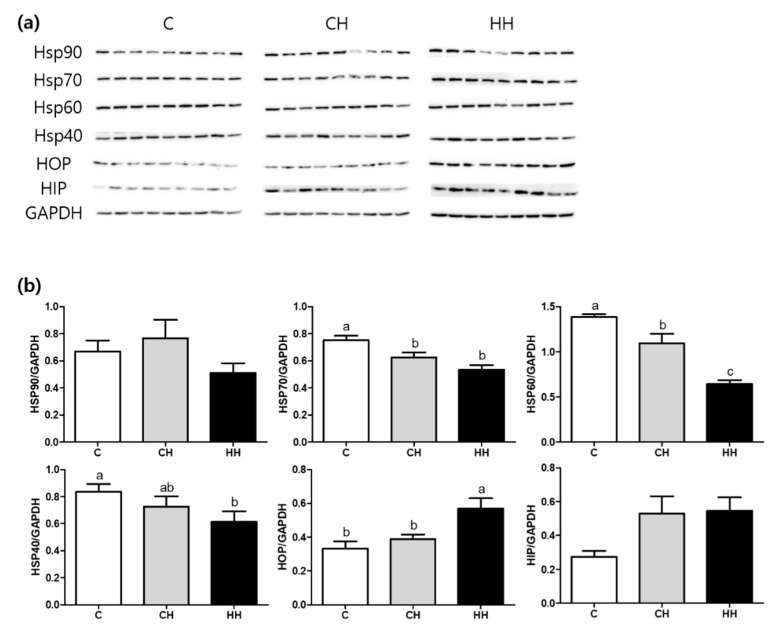
HSPs (heat shock proteins), HOP (hsp70-hsp90 organizing protein), and HIP (hsp70 interacting protein) protein expressions of liver tissue. (**a**) Bands, each line represents a repetition of each birds; (**b**) protein expressions level calculated by GAPDH. C, control; CH, chronic heat-stressed broiler; HH, early and chronic heat-stressed broiler. ^a,b^ Different superscript letters are significantly different (*p* < 0.05).

**Figure 5 animals-09-01022-f005:**
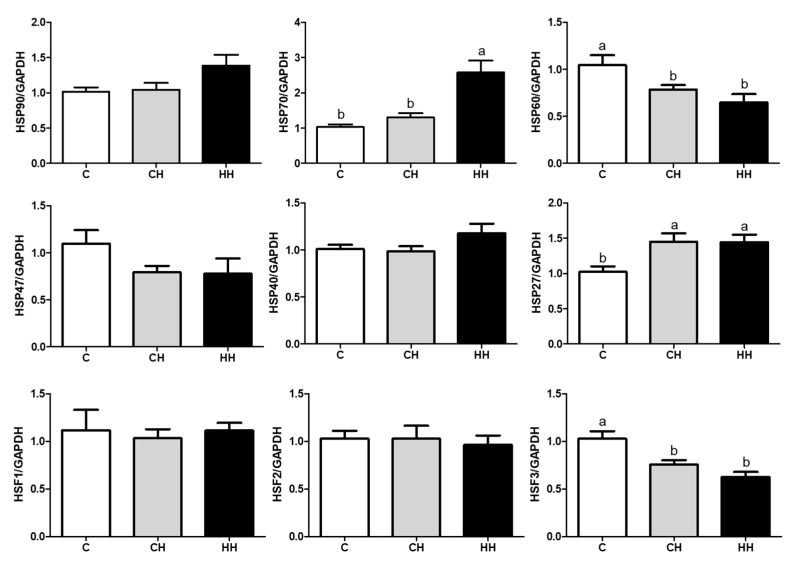
HSPs and HSFs gene expressions of liver. C, control; CH, chronic heat-stressed broiler; HH, early and chronic heat-stressed broiler. ^a,b^ Different superscript letters are significantly different (*p* < 0.05).

**Figure 6 animals-09-01022-f006:**
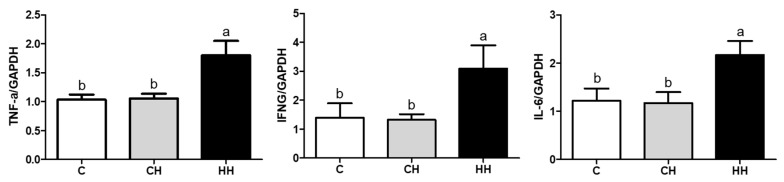
Pro-inflammatory cytokine genes expressions of liver. C, control; CH, chronic heat-stressed broiler; HH, early and chronic heat-stressed broiler. ^a,b^ Different superscript letters are significantly different (*p* < 0.05).

**Table 1 animals-09-01022-t001:** Feed chemical composition of basal diet.

Chemical Composition	Starter ^1^	Finisher ^2^
Crude protein	20.0%	19.0%
Crude fat	4.0%	4.0%
Calcium	0.75%	0.75%
Phosphate	0.70%	0.70%
Crude fiber	6.0%	5.5%
Crude ash	8.0%	8.0%
Met + Cys + MHA ^3^	0.75%	0.65%
ME ^4^	3.00 Mcal/kg	3.05 Mcal/kg

^1^ Starter, 0–20 day; ^2^ Finisher, 21–35 day; ^3^ Met + Cys + MHA, DL-Methionine + Cysteine + DL-Methionine hydroxyl analogue; ^4^ ME, metabolizable energy.

**Table 2 animals-09-01022-t002:** LC-MS/MS condition for dopamine, serotonin, and corticosterone analysis.

Machine	LC-MS/MS, Waters Xevo TQ-S, USA
Column	Synergi Hydro-RP column, 4 μm, 150 × 2 mm; Waters, USA
Injection volume	5 μL
Column temperature	35 °C
Flow rate	0.2 mL/min
Mobile phase	A: 0.1% formic acid in distilled water
	B: 0.1% formic acid in methanol
Gradient program	Initial to 1 min for 0%
	1–4 min for linear increase to 100%
	Hold for 0.5 min
	4.5–5 min for linear decrease to 0% B
	Hold 0% B for 5 min

**Table 3 animals-09-01022-t003:** Target gene primer sequences for qRT-PCR analysis.

Genes	Primer Sequences (5′-3′)	Accession Number	Annealing Temp. (°C)
HSP90	F: TGAAACACTGAGGCAGAAGG	NC_006092.5	62 °C
	R: AAAGCCAGAGGACAGGAGAG		
HSP70	F: GGTAAGCACAAGCGTGACAATGCT	AY143693.1	55 °C
	R: TCAATCTCAATGCTGGCTTGCGTG		
HSP60	F: ATGTGTGGAGCAGCAAGACAGAGA	NM_001012916.1	55 °C
	R: TTCATGAGCTCCCAATCCCAGACA		
HSP47	F: ACTGGCTCATAAGCTCTCCAGCAT	X57157.1	57 °C
	R: TCATCTTGCTGGCCCA GGTCTTTA		
HSP40	F: GGGCATTCAACAGCATAGA	NM_001199325.1	60 °C
	R: TTCACATCCCCAAGTTTAGG		
HSP27	F: ACACGAGGAGAAACAGGATGAG	NM_205290.1	60 °C
	R: ACTGGATGGCTGGCTTGG		
HSF1	F: AAGTCACCAGCGTGTCCAG	NM_001305256.1	60 °C
	R: GCCTCGTTCTCATGCTTCA		
HSF2	F: TACTGCATTTCCGCTGCTC	NM_001167764.2	60 °C
	R: AGGGGTTTGTCCACAGAGG		
HSF3	F: ACGACGTCATCTGCTGGAG	NM_001305041.1	60 °C
	R: TTGAGCTGTCGGATGAAGC		
TNF-α	F: AGATGGGAAGGGAATGAACC	NM_204267.1	60 °C
	R: GACGTGTCACGATCATCTGG		
IFNG	F: TGAACTGAGCCATCACCAAG	NM_205149.1	60 °C
	R: AGGTCCACCGTCAGCTACAT		
IL-6	F: CTCCTCGCCAATCTGAAGTC	NM_204628	60 °C
	R: CCCTCACGGTCTTCTCCATA		
GAPDH	F: AGAACATCATCCCAGCGT	K01458	60 °C
	R: AGCCTTCACTACCCTCTTG		

**Table 4 animals-09-01022-t004:** Growth performance of broiler (means ± S.E).

Factors	Control	Chronic Heat		*p*-Value
	C	CH	HH	
Initial body weight (g/bird)	33.02 ± 0.62	32.48 ± 0.7	32.55 ± 0.58	0.8162
Body weight (g/bird)				
7 days	118.12 ± 3.65	120.45 ± 3.63	114.98 ± 3.79	0.0517
21 days	814.45 ± 17.66	797.05 ± 17.51	777.71 ± 21.22	0.4062
35 days	2029.18 ± 13.51 ^a^	1812.75 ± 11.94 ^b^	1834.58 ± 8.48 ^b^	<0.0001
BDG ^1^ (g/bird)				
0–7 days	12.16 ± 0.53	12.57 ± 0.5	11.78 ± 0.54	0.5613
8–21 days	99.48 ± 2.12	96.66 ± 2.16	94.68 ± 2.82	0.3850
22–35 days	173.53 ± 2.44 ^a^	145.1 ± 2.88 ^b^	150.98 ± 2.95 ^b^	<0.0001
Feed intake (g/bird)				
0–7 days	92.2 ± 1.81	89.1 ± 1.39	87.02 ± 1.11	0.0915
8–21 days	927.16 ± 13.82	906.16 ± 15.7	908.11 ± 22.9	0.6693
22–35 days	2266.32 ± 23.67 ^a^	1994.9 ± 39.2 ^b^	1989.77 ± 20.74 ^b^	0.0001
FCR ^2^				
0–7 days	1.1 ± 0	1.1 ± 0.04	1.03 ± 0.03	0.1410
8–21 days	1.38 ± 0.03	1.35 ± 0.03	1.4 ± 0	0.3227
22–35 days	1.85 ± 0.03	1.93 ± 0.03	1.88 ± 0.03	0.1439

^1^ BDG, body weight daily gain; ^2^ FCR, feed conversion ratio. ^a,b^ Different superscript letters are significantly different (*p* < 0.05).

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
