# Peer review of "Heat Treatment at an Early Age Has Effects on the Resistance to Chronic Heat Stress on Broilers"

_animals, 2019, doi:10.3390/ani9121022_

Round 1

Reviewer 1 Report

Manuscript ID: animals-634246
 Title: Heat treatment at an early age effects on chronic heat stress resistance of broilers

General comment

The Authors investigated the effects of early heat conditioning on growth performance, liver enzymes, as GOT and GPT, neuro-hormones, stress hormones, and gene expression in young broiler chickens under chronic heat stress following early heat exposure.

The topic is interesting, the experimental design is appropriate.

Specific comments are below reported.

Title: I suggest to change it. E.g. “Heat treatment at an early age has effects on the resistance (or resilience) to chronic heat stress on broilers.

Simple Summary

Report that the study regards the broilers…

Line 16: delete “As a results,”

Abstract

Line 23: “ under general temperature” !!!

Lines 23-26: include control group

Line 26: explain what “ CH” means.

Introduction

Lines 40-42: It is not very clear. I suggest to short, e.g: “Heat stress (HS), caused by climate change, negatively affects animal performance, especially chickens during the summer season.”

or “Climate change associated heat stress ………”

Line 44: "resilience" could be better than "resistance"”

Line 52: “Arjona, Denbow and Weaver Jr” change with “Arjona et al.”

M&M

What was the lighting program used?

Where the birds were reared in collective cages or on the floor? If in cages, please include the dimensions (length × width × height). If on the floor the square meters/each birds.

Line 82: “degree” …. Temperature!

Line 90: Were feed intake and body weight measured for each replication? Add details.

Lines 90-91: “Feed intake and body weight were measured on days 0, 7, 14, 90 21, 28, and 35.” After add “Body weight daily gain (BDG) and feed conversion ratio (FCR) were calculated on ………….”

Line 91: Considering that the number of replicate was 4, why were 10 birds selected for each experimental group? How many birds for each replica? It would have been more appropriate to balance the number of chickens (8 or 12), 2 per each replica.

Results

Line 157: Include “(p < 0.05)”

Line 158: Include “(p > 0.05)”

Line 163 and line 172: I suggest to change “decreased” with “lower”; in addition, include p value.

Line 172-173: I suggest to change the “However, no significant differences were detected between the control and the CH group, and the CH and the HH group.” With: “Intermediate value was found in CH group (p > 0.05)”.

Line 185: HSP47 was not reported in Figure 3!

Lines 196-197: HSP27 of CH group was higher compared with the control. Add it

Lines 196-200. Add p values

Discussion

Line 214: Yalcin et al.

Lines 211-212: Comments: Dampened growth performance during heat is not surprising. Growth rate in broiler chickens mainly depends on the level of feed consumed. It can be explained by the fact that digestion, absorption, and metabolism of the nutrients increase metabolic heat production. In high ambient temperature, animal needs to minimize the production rate of metabolic heat and therefore it reduces FI. (see Slawinska et al. 2019  Impact of galactooligosaccharides delivered in ovo on mitigating negative effects of heat stress on performance and welfare of broilers)

Line 217-218: “The CH (before marketing, 2 weeks) group showed a lower body weight gain than the control group, and the HH treatment group showed no effect.” This sentence is not clear for me, rephrased it.

Line 219: Zaboli et al

Line 225: Arjona et al

Line 237: delete the comma after  “Ma”

Line 238: “Our previous results …” What do you mean for “previous results”?

Line 264: replace “studies” with “study”

Line 267. “Previous studies”  Which studies?

Lines 286-294:

I suggest to delete “In conclusion, many studies have been conducted to increase the adaptability to HS following early heat treatment. However, most of them measured growth performance, physiology and oxidative reactions, and reported inconsistent results. We “

I suggest e.g.: “In this study was determined the growth performance, …..”

Tables

Check the punctuation for all tables and figures. Change p in p

Table 4

Include the ANOVA significance P value for each parameters

1.4±… Include Standard error

Include footnotes for CH, HH and for the small letters of the significance values (a and b)

Line 164: p in italic “p”

Author Response

Thank you for taking the time to review our paper.

And, thanks to you, our papers have been upgraded. The response to the comment is attached to the file, along with the revised paper.

Reviewer: 1

General comment

The Authors investigated the effects of early heat conditioning on growth performance, liver enzymes, as GOT and GPT, neuro-hormones, stress hormones, and gene expression in young broiler chickens under chronic heat stress following early heat exposure.

The topic is interesting, the experimental design is appropriate.

Specific comments are below reported.

Title: I suggest to change it. E.g. “Heat treatment at an early age has effects on the resistance (or resilience) to chronic heat stress on broilers.

Answer: We have change the title according to your suggestion.

Simple Summary

Report that the study regards the broilers…

Line 16: delete “As a results,”

Answer: We deleted and add ‘Report that the study regards the broilers’ in the sentence.

Abstract

Line 23: “ under general temperature” !!!

Answer: We have changed that ‘under general temperature’ to ‘under conventional temperature’.

Lines 23-26: include control group

Answer: We explained about control group that ‘without any heat exposure’.

Line 26: explain what “ CH” means.

Answer: We add the explanation of three groups.

Introduction

Lines 40-42: It is not very clear. I suggest to short, e.g: “Heat stress (HS), caused by climate change, negatively affects animal performance, especially chickens during the summer season.”

or “Climate change associated heat stress ………”

Answer: We have modified that sentence according to your suggestion.

Line 44: "resilience" could be better than "resistance"”

Answer: We have changed that word to ‘resilience’.

Line 52: “Arjona, Denbow and Weaver Jr” change with “Arjona et al.”

Answer: We have modified it to your suggestion.

M&M

What was the lighting program used?

Answer: We have added the lighting program within materials and methods.

Where the birds were reared in collective cages or on the floor? If in cages, please include the dimensions (length × width × height). If on the floor the square meters/each birds.

Answer: We have attached the information of cages in materials and method.

Line 82: “degree” …. Temperature!

Answer: We have changed ‘degree’ to ‘temperature’.

Line 90: Were feed intake and body weight measured for each replication? Add details.

Answer: We have add more information about feed intake and body weight within the manuscript.

Lines 90-91: “Feed intake and body weight were measured on days 0, 7, 14, 90 21, 28, and 35.” After add “Body weight daily gain (BDG) and feed conversion ratio (FCR) were calculated on ………….”

Answer: We have suggested that BDG and FCR information.

Line 91: Considering that the number of replicate was 4, why were 10 birds selected for each experimental group? How many birds for each replica? It would have been more appropriate to balance the number of chickens (8 or 12), 2 per each replica.

Answer: Thank you for writing your comments with great detail. We tried to 8 birds for sampling, but seemed too few number, therefore we planned the sampling with 10 birds. Thus, 2~3 chickens were randomly selected from each cages (4 replicated). Thank you again to your comment, we found it’s important to sampling same number of each replica. If we proceed to the next experiment, we will do a balanced sampling as you suggest.

Results

Line 157: Include “(p < 0.05)”

Answer: We have included ‘(p<0.05)’.

Line 158: Include “(p > 0.05)”

Answer: We have included ‘(p>0.05)’.

Line 163 and line 172: I suggest to change “decreased” with “lower”; in addition, include p value.

Answer: We have changed that words and included p value.

Line 172-173: I suggest to change the “However, no significant differences were detected between the control and the CH group, and the CH and the HH group.” With: “Intermediate value was found in CH group (p > 0.05)”.

Answer: We have changed that sentence according to your suggestion.

Line 185: HSP47 was not reported in Figure 3!

Answer: We have deleted that ‘HSP47’ within that sentence.

Lines 196-197: HSP27 of CH group was higher compared with the control. Add it

Answer: We have added that sentence.

Lines 196-200. Add p values

Answer: We have added p values within that part.

Discussion

Line 214: Yalcin et al.

Answer: We have modified that citation.

Lines 211-212: Comments: Dampened growth performance during heat is not surprising. Growth rate in broiler chickens mainly depends on the level of feed consumed. It can be explained by the fact that digestion, absorption, and metabolism of the nutrients increase metabolic heat production. In high ambient temperature, animal needs to minimize the production rate of metabolic heat and therefore it reduces FI. (see Slawinska et al. 2019  Impact of galactooligosaccharides delivered in ovo on mitigating negative effects of heat stress on performance and welfare of broilers)

Answer: Thank you for referring recent article “Slawinska et al 2019”. We have included the reference in the revised manuscript.

Line 217-218: “The CH (before marketing, 2 weeks) group showed a lower body weight gain than the control group, and the HH treatment group showed no effect.” This sentence is not clear for me, rephrased it.

Answer: We have rephrased that sentence to more easily understand.

Line 219: Zaboli et al

Answer: We have changed the citation.

Line 225: Arjona et al

Answer: We have changed the citation.

Line 237: delete the comma after  “Ma”

Answer: We have deleted the comma after ‘Ma’.

Line 238: “Our previous results …” What do you mean for “previous results”?

Answer: We have deleted ‘previous’ within that sentence.

Line 264: replace “studies” with “study”

Answer: We have replace ‘studies’ to ‘study’.

Line 267. “Previous studies”  Which studies?

Answer: Previous studies means Zulkifli et al and Tamzil et al. therefore we modified ‘similarly’ to ‘which’.

Lines 286-294:

I suggest to delete “In conclusion, many studies have been conducted to increase the adaptability to HS following early heat treatment. However, most of them measured growth performance, physiology and oxidative reactions, and reported inconsistent results. We “ I suggest e.g.: “In this study was determined the growth performance, …..”

Answer: We deleted and modified according to your suggestion. 

Tables

Check the punctuation for all tables and figures. Change p in p

Answer: We have changed the punctuation for all tables and figures.

Table 4

Include the ANOVA significance P value for each parameters

Answer: We have attached p-values for each parameter.

1.4±… Include Standard error

Answer: We included standard error.

Include footnotes for CH, HH and for the small letters of the significance values (a and b)

Answer: We have included footnotes for significance values.

Line 164: p in italic “p”

Answer: We have changed all ‘p’ in italic type.

Reviewer 2 Report

In this manuscript "Heat treatment at an early age effects on chronic heat stress resistance of broilers” by DaRae Kang and co-authors presented their work on the effects of heat treatment at an early age on chronic heat stress resistance in broiler chicken.

There are a few points that have to be addressed and modify the manuscript before publishing: Major revision.

Authors are advised to rewrite the abstract—Precise and which can be better understandable by the readers. What is CH and HH stress groups? Elaborate CH, HH, HSPs and HSFs in the abstract. Abbreviations are elaborated at its first appearance in the manuscript. HSF, HOP, and HIP. The last paragraph of the abstract is very confusing. Authors are suggested to rectify the confusion. In Line 29 authors wrote that HSP gene expression was down regulated but at line 32 they have indicated that HSP70 and HSP27 gene expression was up-regulated. Specify which HSP genes were down-regulated and which are up-regulated. Keywords are not in alphabetical order. Authors are suggested to include few more references at line 48, as the line states that –most of the previous studies reported that…………………oxidative damage markers and hormonal changes. It would be better if the authors represent the animal treatments in a graphical / diagrammatic representation. References should be provided for the method of determination of serum hormones levels, western blot analysis of HSPs, HOP, and HIP. Table 4: Footnote of the growth performance of broiler in table 4 was not represented clearly. Authors are suggested to represent the statistical significance as footnote in Table 4 and give P values. For better understandability, figure 1, figure 2, and figure 3 should contain sub grouping such as figure 1A for SGOT and figure 1B for SGPT and so on.. Figure 3(a): Authors have presented lot many bands in each group for each single protein. What does each band represents and how have the authors compared or calculated the statistical significance between the groups. Results of HSP47: Bands and gene expression of HSP47 are not presented in the manuscript either in text or as figure. Why HSP27 m-RNA expression was analyzed but not protein expression? The statistical representation of footnote under tables and figures was not up to the mark. Authors are suggested to include the following reference at line 231 after liver damage. Lavanya et al., 2017. Preventive and curative effects of Cocculus hirsutus (Linn.) Diels leaves extract on CCl4 provoked hepatic injury in rats. Egyptian Journal of Basic and Applied Sciences. 4 (4), 264-269. Discussion part is not up to the mark. The results are repeated in the discussion. Authors are recommended to avoid the repetition. Include figures and tables numbers where ever required to discuss them. Authors are suggested to strengthen the importance and relevance of the present results in the discussion section. Authors have repeatedly indicated that many studies in this particular area reported inconsistent results –In what way they mean the previous studies results are inconsistent and what is the percentage of reproducibility of present study? Authors should check the conclusion part: Are the results really supporting the conclusion? The tested /selected parameters are sufficient for the conclusion? Should re-write the conclusion by including future directions. References should be cited by following journal style/format. Need to check for typographical errors, plagiarism, punctuation, and grammar throughout the manuscript.

Author Response

Thank you for taking the time to review our paper.

And, thanks to you, our papers have been upgraded. The response to the comment is attached to the file, along with the revised paper. 

Please wee the attachment.

Reviewer: 2

In this manuscript "Heat treatment at an early age effects on chronic heat stress resistance of broilers” by DaRae Kang and co-authors presented their work on the effects of heat treatment at an early age on chronic heat stress resistance in broiler chicken.

There are a few points that have to be addressed and modify the manuscript before publishing: Major revision.

Authors are advised to rewrite the abstract—Precise and which can be better understandable by the readers.

Answer: We have modified the abstract for better understandable.

What is CH and HH stress groups?

Answer: We have explained about CH and HH groups.

Elaborate CH, HH, HSPs and HSFs in the abstract.

Answer: We elaborated about the abbreviations in the abstract.

Abbreviations are elaborated at its first appearance in the manuscript. HSF, HOP, and HIP.

Answer: We have elaborated all abbreviations at its first appearance in the manuscript.

The last paragraph of the abstract is very confusing. Authors are suggested to rectify the confusion.

Answer: We have modified the abstract for better understandable.

In Line 29 authors wrote that HSP gene expression was down regulated but at line 32 they have indicated that HSP70 and HSP27 gene expression was up-regulated. Specify which HSP genes were down-regulated and which are up-regulated.

Answer: We wrote that line 29 ‘the expressions of HSPs were down-regulated’, which HSPs is proteins. And in line 32 ‘while HSP70 and HSP27 genes were up-regulated’ that HSP70 and HSP27 are genes. Therefore we add ‘protein’ for avoid confusing.

Keywords are not in alphabetical order.

Answer: We sorted the keywords in alphabetical order.

Authors are suggested to include few more references at line 48, as the line states that –most of the previous studies reported that…………………oxidative damage markers and hormonal changes.

Answer: We have appeared the references at the end of sentence.

It would be better if the authors represent the animal treatments in a graphical / diagrammatic representation.

Answer: Thank you for your suggestion about animal treatments. We have made a graphical animal treatment in Figure 1.

References should be provided for the method of determination of serum hormones levels, western blot analysis of HSPs, HOP, and HIP.

Answer: We have suggested the reference about methods.

Table 4: Footnote of the growth performance of broiler in table 4 was not represented clearly. Authors are suggested to represent the statistical significance as footnote in Table 4 and give P values.

Answer: We have added footnote of table4 about statistical significance.

For better understandability, figure 1, figure 2, and figure 3 should contain sub grouping such as figure 1A for SGOT and figure 1B for SGPT and so on.

Answer: We have modified Figure1 and 2 according to your suggestion for better understandability. However, in the case of Figure 3, it was more confusing, so we kept this format. We hope you understand about this part.

Figure 3(a): Authors have presented lot many bands in each group for each single protein. What does each band represents and how have the authors compared or calculated the statistical significance between the groups.

Answer: Each line represented individual birds of each group. We have modified about bands and calculated method in materials and method, and figure 3 footnote.

Results of HSP47: Bands and gene expression of HSP47 are not presented in the manuscript either in text or as figure.

Answer: Thank you for check our mistake. We have deleted about HSP47, and modified that sentence.

Why HSP27 m-RNA expression was analyzed but not protein expression?

Answer: We tried to detect HSP27 protein, but hsp27 protein was not detected. Thus, we

The statistical representation of footnote under tables and figures was not up to the mark.

Answer: We have added footnote under tables and figures.

Authors are suggested to include the following reference at line 231 after liver damage. Lavanya et al., 2017.Preventive and curative effects of Cocculus hirsutus (Linn.) Diels leaves extract on CCl4 provoked hepatic injury in rats. Egyptian Journal of Basic and Applied Sciences. 4 (4), 264-269.

Answer: Thank you for your suggestion. We have put that reference after liver damage.

Discussion part is not up to the mark.

Answer: We have added more references in discussion part for supporting the results.

The results are repeated in the discussion. Authors are recommended to avoid the repetition.

Answer: We have removed the repeated results in the discussion.

Include figures and tables numbers where ever required to discuss them.

Answer: We have included the figures and tables numbers within discussion part.

Authors are suggested to strengthen the importance and relevance of the present results in the discussion section.

Answer: Thank you for your suggestion, we have included in conclusion part.

Authors have repeatedly indicated that many studies in this particular area reported inconsistent results –In what way they mean the previous studies results are inconsistent and what is the percentage of reproducibility of present study?

Answer: The inconsistency of previous studies is that the results for early heat exposure are either positive or negative. Although early heat exposure studies have been conducted before, reproducibility is difficult to explain as percentages because the area where the experiment is conducted, the breed of the broiler, and other environmental conditions vary from study.

Authors should check the conclusion part: Are the results really supporting the conclusion? The tested /selected parameters are sufficient for the conclusion? Should re-write the conclusion by including future directions.

Answer: We have re-written the results and future directions within conclusion part for more supporting the conclusion.

References should be cited by following journal style/format.

Answer: We have re-checked and modified according to journal style/format.

Need to check for typographical errors, plagiarism, punctuation, and grammar throughout the manuscript.

Answer: We have checked for typographical errors, plagiarism, punctuation, and grammar throughout the manuscript.

Round 2

Reviewer 1 Report

I suggest to change the sentence “Body weight daily gain (BDG) and feed conversion ratio (FCR) were calculated using weekly feed intake and body weight.”  in “Cumulative feed intake and feed conversion ratio (FCR) were calculated on a cage basis.”

Table 4

Title: I suggest to change “Growth performance of broiler.” With “Growth performance of broiler (means ± S.E).

Author Response

Thank you for taking the time to review our paper.

And, thanks to you, our papers have been more upgraded by your suggestions. The response to the comment is attached to the file, along with the revised paper.

Reviewer 1.

I suggest to change the sentence “Body weight daily gain (BDG) and feed conversion ratio (FCR) were calculated using weekly feed intake and body weight.”  in “Cumulative feed intake and feed conversion ratio (FCR) were calculated on a cage basis.”

Answer: We have changed that sentence according to your comment.

Table 4

Title: I suggest to change “Growth performance of broiler.” With “Growth performance of broiler (means ± S.E).

Answer: We have changed the title of Table 4 according to your suggestion.

Reviewer 2 Report

In this manuscript "Heat treatment at an early age effects on chronic heat stress resistance of broilers” by DaRae Kang and co-authors presented their work on the effects of heat treatment at an early age on chronic heat stress resistance in broiler chicken. I think the manuscript has been improved. Still there are few points that have to be addressed and modify in the manuscript before publishing.

Page 4---"Determination of serum dopamine,serotonin, and corticosterone levels"--change first two sentences as dopamine, serotonin, and corticosterone levels in the serum were analyzed via LC-MS/MS according to the slightly modified method of Marwah et al., specify year [16].

References should be cited according to the journal style/format.

Lot of grammatical mistakes were found. Authors are suggested to go through the manuscript carefully and rectify the grammatical mistakes.

Need to check the manuscript once again for typographical errors, plagiarism, and punctuation.

Author Response

Thank you for taking the time to review our paper.

And, thanks to you, our papers have been more upgraded by your suggestions. The response to the comment is attached to the file, along with the revised paper.

Reviewer 2.

In this manuscript "Heat treatment at an early age effects on chronic heat stress resistance of broilers” by DaRae Kang and co-authors presented their work on the effects of heat treatment at an early age on chronic heat stress resistance in broiler chicken. I think the manuscript has been improved. Still there are few points that have to be addressed and modify in the manuscript before publishing.

Page 4---"Determination of serum dopamine,serotonin, and corticosterone levels"--change first two sentences as dopamine, serotonin, and corticosterone levels in the serum were analyzed via LC-MS/MS according to the slightly modified method of Marwah et al., specify year [16].

Answer: We have changed the sentences according to your suggestion.

References should be cited according to the journal style/format.

Answer: We have corrected the References according to journal style/format.

Lot of grammatical mistakes were found. Authors are suggested to go through the manuscript carefully and rectify the grammatical mistakes.

Answer: We had one time proofreading before submitting this paper via HARISCO English editing company. Since we only got two days to review (18 November to 20 November for revision period), it was unable to proceed with corrections by editing company. Therefore, this time, grammatical mistakes were corrected through native English speaker in our department.

Need to check the manuscript once again for typographical errors, plagiarism, and punctuation.

Answer: We have checked again the manuscript for typographical errors, plagiarism, and punctuation.